# Endoglin Promotes Myofibroblast Differentiation and Extracellular Matrix Production in Diabetic Nephropathy

**DOI:** 10.3390/ijms21207713

**Published:** 2020-10-18

**Authors:** Tessa Gerrits, Malu Zandbergen, Ron Wolterbeek, Jan A. Bruijn, Hans J. Baelde, Marion Scharpfenecker

**Affiliations:** 1Department of Pathology, Leiden University Medical Centre, 2333 ZA Leiden, The Netherlands; j.a.bruijn@lumc.nl (J.A.B.); j.j.baelde@lumc.nl (H.J.B.); m.scharpfenecker@lumc.nl (M.S.); 2Department of obstetrics and gynecology, The University of Chicago, Chicago, IL 60637, USA; mzandbergen@bsd.uchicago.edu; 3Department of Biomedical Data Sciences, Leiden University Medical Centre, 2333 ZA Leiden, The Netherlands; r.wolterbeek@lumc.nl

**Keywords:** diabetic nephropathy, endoglin, fibroblast, interstitial fibrosis, TGF-β

## Abstract

Diabetic nephropathy (DN) is a complication of diabetes mellitus that can lead to proteinuria and a progressive decline in renal function. Endoglin, a co-receptor of TGF-β, is known primarily for regulating endothelial cell function; however, endoglin is also associated with hepatic, cardiac, and intestinal fibrosis. This study investigates whether endoglin contributes to the development of interstitial fibrosis in DN. Kidney autopsy material from 80 diabetic patients was stained for endoglin and Sirius Red and scored semi-quantitatively. Interstitial endoglin expression was increased in samples with DN and was correlated with Sirius Red staining (*p* < 0.001). Endoglin expression was also correlated with reduced eGFR (*p* = 0.001), increased creatinine (*p* < 0.01), increased systolic blood pressure (*p* < 0.05), hypertension (*p* < 0.05), and higher IFTA scores (*p* < 0.001). Biopsy samples from DN patients were also co-immunostained for endoglin together with CD31, CD68, vimentin, or α-SMA Endoglin co-localized with both the endothelial marker CD31 and the myofibroblast marker α-SMA. Finally, we used shRNA to knockdown endoglin expression in a human kidney fibroblast cell line. We found that TGF-β1 stimulation upregulated *SERPINE1*, *CTGF*, and *ACTA2* mRNA and α-SMA protein, and that these effects were significantly reduced in fibroblasts after endoglin knockdown. Taken together, these data suggest that endoglin plays a role in the pathogenesis of interstitial fibrosis in DN.

## 1. Introduction

Approximately 40% of patients with diabetes develop diabetic nephropathy (DN) [1], the leading cause of end-stage renal disease in the Western world; moreover, the incidence of DN continues to rise [2]. The clinical symptoms associated with DN include microalbuminuria, proteinuria, hypertension, and reduced glomerular filtration rate [1,3]. Histologically, DN is characterized by an expansion of the extracellular matrix (ECM) in the glomerulus and—in advanced stages—interstitial fibrosis. Tubulointerstitial changes, including fibrosis, can lead to a loss of kidney function and organ failure [4,5].

The cytokine TGF-β (transforming growth factor-beta) plays an important role in developing fibrosis, including renal fibrosis [6]. TGF-β induces the production of ECM proteins via connective tissue growth factor (*CTGF*) and/or plasminogen activator inhibitor-1 (PAI-1, encoded by the *SERPINE1* gene) [7,8,9] and induces the differentiation of fibroblasts into myofibroblasts that express alpha smooth muscle actin (α-SMA) [10,11]. These myofibroblasts represent an activated fibroblast phenotype with the capacity to synthesize large amounts of ECM proteins [12,13].

Endoglin (also known as CD105) is a membrane glycoprotein that interacts with TGF-β receptors and is expressed in a variety of cell types, including endothelial cells [14], monocytes/macrophages [15], smooth muscle cells [16], and fibroblasts [17]. Evidence suggests that endoglin promotes the development of fibrosis. For example, in various organs, such as the liver, heart, and intestine, endoglin expression is higher in fibroblasts in fibrotic tissue compared to non-fibrotic tissue [18,19,20]. In cardiac fibroblasts, angiotensin II increases the expression of endoglin at both the mRNA and protein levels and induces the production of type I collagen, a process that can be blocked using an endoglin antibody [21,22]. Moreover, endoglin is upregulated in the renal interstitium in several animal models of renal fibrosis, including unilateral ureter obstruction (UUO), ischemia-reperfusion injury, and radiation-induced nephropathy [23,24,25]. In line with the high endoglin expression described in fibrotic tissue, overexpressing endoglin in the UUO model aggravates renal fibrosis [26]. Supporting the notion of endoglin as a pro-fibrotic molecule, the reduced endoglin levels in heterozygous endoglin-knockout (ENG+/-) mice reduce cardiac fibrosis in a model of pressure overload-induced heart failure [27]. Accordingly, compared to wild-type mice, endoglin haploinsufficient mice have reduced radiation-induced renal fibrosis, as well as reduced numbers of myofibroblasts [28]. 

In this study, we investigated whether endoglin also plays a role in the pathogenesis of renal fibrosis in DN. We first measured renal endoglin expression in autopsy samples obtained from diabetic patients with and without DN and correlated the results with the degree of interstitial fibrosis and several clinical parameters. We also analyzed endoglin expression in whole kidney lysates from a biopsy cohort. We then determined which cell types express endoglin in the interstitium using biopsy material obtained from patients with DN. Lastly, we investigated the effect of reducing endoglin expression on myofibroblast differentiation and ECM production in kidney fibroblasts using an in vitro approach. Our results support a role for endoglin in developing renal fibrosis in human DN.

## 2. Results

### 2.1. Patients with Diabetic Nephropathy Have Increased Endoglin Expression that is Correlated with Various Clinical Parameters

First, we measured endoglin levels in the interstitium of patients with DN and patients without DN. In the autopsy cohort, semi-quantitative analysis of endoglin-stained sections revealed that the patients with histologically confirmed DN had significantly higher levels of endoglin in the interstitium compared to diabetic controls without DN (*p* < 0.05) (Figure 1A). Moreover, among the patients with DN, the endoglin-positive interstitial area was positively correlated with the fibrotic index measured based on Sirius Red staining (*p* < 0.001) (Figure 1B). With respect to the clinical data, the endoglin-positive interstitial area negatively correlated with estimated glomerular filtration rate (eGFR) (*p* < 0.02), positively correlated with the presence of hypertension (*p* < 0.05), Interstitial fibrosis and tubular atrophy (IFTA) score (*p* < 0.001), and serum creatinine concentration (*p* < 0.001) (Figure 1C). Appendix A describes the characteristics of DN patients within the three groups with different endoglin expression. 

To confirm the findings obtained in the autopsy samples, we also measured endoglin levels in eleven kidney biopsies from patients with DN; as a control, we stained tumor-free kidney samples from seven non-diabetic patients who underwent a renal tumor resection. Endoglin was expressed in the endothelial cells in both groups; however, endoglin was also expressed by cells in the interstitium (Figure 2A,B). Digital image analysis revealed that the endoglin-positive interstitial area was significantly larger in patients with DN compared to controls (*p* < 0.002) (Figure 2E). Moreover, the Sirius Red positive area was larger in patients with DN compared to controls (*p* < 0.001) (Figure 2C,D,F), which indicates more collagen deposition. 

Endoglin expression was also investigated in whole kidney lysates of 30 DN patients and 12 healthy controls. Gene expression analysis showed that mRNA of endoglin was 1.86-fold upregulated in patients with DN compared to controls (*p* < 0.05) (Figure 2G).

### 2.2. Endoglin Is Expressed by Interstitial Myofibroblasts

Next, we identified which cell type in the interstitium expresses endoglin by performing double-immunofluorescence staining of kidney biopsies from patients with DN and controls. Our analysis showed that part of the endoglin positivity co-localized with the endothelial cell marker CD31, confirming endoglin expression in blood vessels. Yet, we also found that in patients with DN, endoglin expression co-localized with the fibroblast marker vimentin and with the myofibroblast marker α-SMA (Figure 3). These data indicate that interstitial endoglin is mainly expressed in (myo) fibroblasts.

### 2.3. Endoglin Knockdown Reduces the Expression of TGF-β Downstream Targets and Inhibits Fibroblast Differentiation

Next, we analyzed the effect of reducing endoglin expression on fibroblast differentiation and ECM production in TK173 cells, a human kidney fibroblast cell line. To knockdown endoglin expression, cells were transduced with a lentiviral construct expressing a small hairpin (sh)RNA to target *ENG* mRNA (*ENG^KD^* fibroblasts); control cells were transduced with a virus expressing an empty vector control (*ENG^ctrl^* fibroblasts). This approach caused a 60% reduction in endoglin mRNA levels (Figure 4A) and a 40% reduction in endoglin protein levels (Figure 4B) in *ENG^KD^* fibroblasts compared to control cells. We considered a knockdown of ~50% sufficient, as a higher reduction of endoglin levels in fibroblasts has been described to lead to a senescent phenotype and subsequent cell detachment and death [29].

We performed an α-SMA staining on fibroblasts cultured on coverslips. The intensity of the α-SMA staining seemed to be slightly lower in the *ENG^KD^* compared to the *ENG^ctrl^* cells (Figure 5A,B). To quantitatively assess changes in α-SMA, mRNA and protein levels were analyzed. To this end, *ENG^KD^* and *ENG^ctrl^* fibroblasts were stimulated with TGF-β1, and gene expression of *ACTA2* (which encodes α-SMA) and α-SMA protein were measured using qPCR and western blot analysis, respectively. TGF-β1 significantly increased both *ACTA2* mRNA and α-SMA protein levels in *ENG^ctrl^* fibroblasts. In contrast, TGF-β1 had no significant effect on either *ACTA2* mRNA or α-SMA protein levels in *ENG^KD^* fibroblasts (Figure 5C,D).

Finally, we found that stimulating *ENG^ctrl^* fibroblasts with TGF-β1 significantly upregulated the TGF-β downstream pro-fibrotic target genes *CTGF* and *SERPINE1* (Figure 6A,B). In contrast, although TGF-β1 also upregulated both *CTGF* and *SERPINE1* in *ENG^KD^* fibroblasts, the effect was significantly lower compared to *ENG^ctrl^* fibroblasts (Figure 6A,B). 

### 2.4. The Effect of Endoglin Knockdown on Myofibroblast Differentiation and ECM Production Cannot be Explained by Differences in Canonical TGF-β Signaling. 

We performed a pSmad1/5/9 and pSmad2/3 western blot to investigate whether the effect of endoglin knockdown on α-SMA and ECM production is mediated through the canonical TGF-β pathway. TGF-β stimulation significantly increased the phosphorylation of Smad1/5/9 by approximately 2× (Figure 7A), and the phosphorylation of Smad2/3 by approximately 10× (Figure 7B). No differences in Smad phosphorylation were seen between the *ENG^ctrl^* and *ENG^KD^* fibroblasts (Figure 7A,B).

## 3. Discussion

Here, we report that endoglin levels are increased in the interstitium of diabetic patients with DN compared to diabetic patients without DN. We show that the level of interstitial endoglin is positively correlated with the extent of interstitial fibrosis in patients with DN. In addition, we found that endoglin expression is correlated with several clinical parameters, including eGFR, hypertension, IFTA index score, and serum creatinine level. Importantly, our analysis revealed that (myo)fibroblasts are the endoglin-producing cells in the interstitium. Finally, our in vitro experiments indicate that knocking down endoglin expression in kidney fibroblasts reduces the TGF-β-induced fibroblast-to-myofibroblast differentiation and the production of pro-fibrotic TGF-β downstream target genes. 

So far, only a few studies have investigated endoglin expression in human fibrotic disease. Endoglin upregulation has been reported in myocardial fibrosis [22], in the skin of patients with scleroderma [30], and in intestinal fibroblasts in patients with Crohn’s Disease [20]. To our knowledge, only one study has analyzed a possible link between endoglin in the interstitium of the kidney and the extent of chronic histological damage. This study by Roy-Chaudhury et al., from 1997, investigated biopsies of patients with chronic progressive renal disease, including one patient with DN, and found a weak, albeit significant, correlation between the extent of interstitial endoglin staining and chronic histological damage [31]. 

We now show in two cohorts of patients with DN that the endoglin-positive area was significantly more extensive in patients with DN than in diabetic patients without DN. Moreover, we found a correlation between the level of interstitial endoglin and the degree of interstitial fibrosis. Assuming that interstitial endoglin might play a role in interstitial fibrosis and knowing that there is a link between interstitial fibrosis and decreased renal function [32,33], we analyzed whether there is a link between the endoglin positive area and the clinical data of the patient cohort. We found an inverse correlation between interstitial endoglin levels and renal function in patients with DN, which further supports the potential role of endoglin in renal fibrosis in DN. 

The notion that endoglin might play a role in renal fibrosis is supported by findings from several in vivo studies. For example, Prieto et al. found that endoglin expression was upregulated in the interstitium of the ligated kidney compared to the non-ligated control kidney in a mouse model of UUO, which is characterized by progressive renal fibrosis [25]. Consistent with these findings, Oujo et al. subsequently reported that transgenic mice that ubiquitously overexpress human endoglin have increased renal fibrosis following UUO [26]. Moreover, Docherty et al. described that endoglin is expressed in fibrotic and inflamed areas of the interstitium in a mouse model of renal ischemia-reperfusion [23]. In the same study, the authors also found that renal ischemia-reperfusion has a less severe effect in heterozygous ENG-knockout (ENG+/-) mice compared to wild-type mice with respect to acute tubular necrosis and increased creatinine levels [23]. Similarly, in a radiation injury model of chronic renal fibrosis, it was shown that ENG+/– mice develop less severe inflammation and fibrosis and have improved renal function compared to wild-type littermates [28,34,35]. These findings are in contrast to a study by Rodríguez-Peña et al., who found no difference in UUO-induced kidney damage between endoglin haplo-insufficient mice and wild-type mice [24]. Taken together, the majority of animal studies suggest that endoglin promotes the development of renal fibrosis. 

The double-immunofluorescence stainings on human renal tissue suggest that endoglin present in the interstitium is mainly, however not exclusively, expressed by myofibroblasts, which have an activated fibroblast phenotype, are contractile, still express vimentin, due to differentiation now also express α-SMA, and can produce large amounts of ECM proteins [36]. The accumulation of ECM proteins in the kidney is associated with a loss of tissue structure and scarring [37]. Increased numbers of myofibroblasts have been observed in the kidneys of animal models of DN. In the UUO model, kidney fibrosis is associated with a significant accumulation of α-SMA‒expressing myofibroblasts in the interstitium [38,39]. In KKAy mice, a mouse model for type 2 diabetes, myofibroblast transdifferentiation markers are expressed in the kidneys at 16 weeks of age, followed by tubulointerstitial fibrosis at later stages [40]. Myofibroblast transdifferentiation markers, as well as matrix accumulation, have also been observed in the tubulointerstitium of obese Zucker (fa/fa) diabetic rats [41]. Based on these studies, our findings suggest that the accumulation of endoglin-expressing myofibroblasts in the interstitium of patients with DN could contribute to the formation of interstitial fibrosis. 

In addition to myofibroblasts, positive endoglin staining was also observed in the mesangial area of the glomeruli (Figure 2 and Figure 3). Endoglin expression in mesangial cells has been described earlier by in vitro, showing TGF-β mediated upregulation of endoglin in human and rat glomerular mesangial cells [42,43]. The finding suggest that endoglin might also play a role in mesangial extracellular matrix production, and thereby, in the development of glomerulosclerosis. To investigate whether anti-endoglin therapy could potentially inhibit both interstitial fibrosis and glomerulosclerosis, would therefore be an interesting and relevant subject for future studies. 

To address the putative role of endoglin in myofibroblasts in DN, we focused on two cellular processes related to fibrosis development, namely, the differentiation of fibroblasts into myofibroblasts and the production of ECM proteins. In our experiments, we found that TGF-β-induced expression of the myofibroblast marker α-SMA in cultured kidney fibroblasts was significantly reduced when endoglin expression is knocked down, suggesting that endoglin promotes fibroblast-to-myofibroblast differentiation. Although previous studies have described that TGF-β induces the differentiation of kidney fibroblasts into myofibroblasts [44,45], the role of endoglin in this process has not been studied previously. Nevertheless, our finding that endoglin seems to be required for fibrosis, and may even promote fibrosis by supporting fibroblast-to-myofibroblast differentiation, is supported by a previous report by Meurer et al., who found that overexpressing rat endoglin in immortalized mouse hepatic stellate cells increases the expression of α-SMA [46]. 

Our finding that the TGF-β-induced upregulation of the pro-fibrotic genes *CTGF* and *SERPINE1* is reduced in *ENG^KD^*^+/–^ fibroblasts suggests that endoglin also promotes the production of ECM proteins. This notion is supported by other in vitro studies involving myoblasts showing that endoglin increases TGF-β1-induced collagen I and *CTGF* expression [27,29,47]. Moreover, Morris et al. found that in cells of patients with systemic sclerosis, endoglin has an important pro-fibrotic role [48]. Taken together, these data indicate that endoglin is involved in fibroblast-to-myofibroblast differentiation and promotes TGF-β-induced ECM production, both of which may contribute to the development of renal fibrosis. 

Endoglin modulates TGF-β1 signaling mainly via the downstream pathways Smad1 and 5, but effects on Smad2/3 signaling have also been described [49]. In line with this, it has been shown in several studies that endoglin can enhance TGF-β1-induced Smad1/5 and/or Smad2/3 phosphorylation in scleroderma fibroblasts, liver fibrosis, and cardiac fibrosis [27,48,50].

TGF-β-Smad2/3 signaling is recognized as a major pathway in progressive renal fibrosis [51]. Accordingly, we observed an upregulation of pSmad2/3 after TGF-β stimulation. However, there was no difference in Smad phosphorylation between the *ENG^ctrl^* and *ENG^KD^* fibroblasts. Smad1/5 phosphorylation was also upregulated after TGF-β stimulation, but also here we observed no difference between wild type and endoglin knockdown cells. This could imply that the effect of the endoglin knockdown on α-SMA expression and ECM production is mediated through one of the non-canonical pathways of TGF-β signaling. In the literature, a role for the non-canonical pathway in TGF-β-mediated pro-fibrotic processes has been described. TGF-β-dependent activation of PI3K-Akt can induce morphological myofibroblast-like transformation independent of Smad signaling in a subset of fibroblast cell lines [52]. Moreover, c-Abl-kinase was shown to promote TGF-β-induced pulmonary fibrosis independent of Smad2/3 phosphorylation [53]. Finally, it has been shown that *CTGF* expression in hepatic stellate cells only partly depends on pSmad3, but also on Jak1-Stad3 signaling [54] and that inhibition of JAK-2 can inhibit Bleomycin-induced dermal fibrosis in mice [55].

The immunofluorescent staining on biopsy material showed that endoglin is not only expressed in myofibroblasts, but also in interstitial endothelial cells, as evidenced by co-localization of endoglin with the endothelial cell marker CD31. Previously, we noted upregulation of endoglin and decreased CD31 expression in glomerular blood vessels of patients with DN [56]. We also observed increased expression of VCAM1 in the glomerular endothelium. Downregulation of CD31 and upregulation of VCAM 1 are indicative of endothelial activation, pro-inflammatory changes. In that study, we concluded that increased endoglin levels promote endothelial activation, and thereby, the pro-inflammatory state of the glomerular endothelium in DN. It is possible that the observed endothelial damage might proceed to subsequent vascular rarefaction. In DN, a reduction in peritubular capillary densities shown by reduced CD31 expression in the interstitium has been described [57,58]. In the current study, we did not investigate whether similar changes in these vascular markers also occur in interstitial blood vessels. However, we analyzed *CD31* mRNA levels in whole kidney lysates of the biopsy cohort and found decreased expression of *CD31* in DN patients compared to healthy controls (data not shown). This suggests that similar mechanisms with respect to endothelial activation and subsequent rarefaction might apply to the interstitium as well.

Future studies should focus on testing the feasibility of targeting endoglin as a therapeutic strategy in DN. In this respect, it is interesting to note that several anti-endoglin antibodies are currently in preclinical and clinical development. For example, TRC105, a chimeric Immunoglobulin G (IgG) 1 antibody against human endoglin developed by Tracon Pharmaceuticals in San Diego, CA, is currently being studied as a promising anti-angiogenic agent in combination with VEGF inhibitors in one phase 3 and several phase 2 clinical trials for treating various solid tumor types [59,60]. In addition, TRC205 (also developed by Tracon Pharmaceuticals), an IgG4 antibody against human endoglin, is currently being tested in preclinical studies for its ability to reduce liver fibrosis. In mice, TRC205 significantly decreased carbon tetrachloride-induced hepatic fibrosis and reduced collagen deposition with reduced formation of bridging fibrosis compared to control groups [61]. 

In conclusion, we report that endoglin is upregulated in renal tissue of patients with DN, negatively correlates with renal function, is expressed by myofibroblasts in the interstitium of patients with DN. We also suggest that endoglin plays an important role in renal fibrosis by promoting TGF-β-induced fibroblast-to-myofibroblast differentiation and ECM production. Future studies need to show whether endoglin may serve as a possible therapeutic target for preventing the development of fibrosis in patients with DN, thereby slowing the progression towards end-stage renal disease.

## 4. Materials and Methods 

### 4.1. Autopsy Cohort 

Renal autopsy tissue specimens from diabetic patients were selected from the cohort previously described by Klessens et al. [62] and were retrieved from the pathology archives of the Leiden University Medical Centre (LUMC). The primary inclusion criterion for our analysis was the presence of diabetes in patients who were over the age of 18 years at the time of death. A total of 80 samples from diabetic patients were included from autopsies performed from 1984 to 2005. For our study, only patients with histologically confirmed DN were included in the DN group (*n* = 55); patients without clinical or histological DN and no evidence of kidney insufficiency were included in a diabetic control group (*n* = 25). The renal autopsy samples were scored for the diagnosis of DN by two investigators who were blinded with respect to the patients’ clinical data, and DN was diagnosed in accordance with the established histopathologic classification for DN [63]. 

### 4.2. Biopsy Cohort

Renal biopsy samples from eleven patients with histologically confirmed DN were obtained from the pathology archives of LUMC. As a control group, we also included tumor-free renal resection samples obtained from seven non-diabetic patients who had a kidney tumor in another part of the kidney. All patient samples were collected and handled in accordance with Dutch national ethics guidelines (the Code of Conduct for the Proper Secondary Use of Human Tissue).

Whole kidney lysate from a frozen biopsy cohort described by Baelde et al. (2007) [58] was used to perform a quantitative real-time PCR (qPCR) to determine endoglin levels. These biopsies were obtained from the pathology archives of the LUMC and the Institute for Clinical Pathology, Heidelberg. All patients with DN (*n* = 30) had type II diabetes. The control group (*n* = 12) consisted of the non-affected part of tumor nephrectomy samples.

### 4.3. Clinical Data

Clinical data were retrieved from the patients’ medical records available at the LUMC and/or from the patients’ general practitioners. Approval to collect relevant clinical data from the patients’ general practitioners was obtained from the medical ethics committee of LUMC. The following laboratory results were collected: Serum creatinine eGFR (calculated using the MDRD formula), systolic and diastolic blood pressure, serum hemoglobin, serum cholesterol, and serum HbA1c. For the autopsy cohort, these data were collected retrospectively from the period starting one year before the patient’s death. Data that reflected that the patient’s serum and/or urine levels were stable were included in our analysis. The characteristics of the autopsy cohort are summarized in Table 1.

### 4.4. Immunohistochemistry

Sections (4-µm thick) of paraffin-embedded kidney tissues obtained from the autopsy and biopsy cohorts were cut using a Leica microtome. The sections were subjected to heat-induced antigen retrieval using 10 mM Tris/EDTA (pH 9.0). One slide per patient was stained with a polyclonal goat anti-human endoglin antibody (1:800; R&D Systems, Minneapolis, MN, USA). The sequential slide was stained with Sirius Red (Sigma-Aldrich, Saint Louis, MO, USA). All kidney sections were stained in a single session for immunohistochemical analysis and scanned using a Philips Ultra-Fast Scanner 1.6 RA.

### 4.5. Scoring of Sections and Analysis of Digital Images

In the autopsy cohort, the endoglin-positive area in the entire cortical interstitium was scored semi-quantitatively by two independent observers on images obtained at 100× magnification using the following three categories: Endoglin-positive area (ENG) <25%, ENG 25−50%, and ENG >50%. Consensus regarding the score of each section was reached and used in the statistical analysis. To analyze the Sirius Red staining, three representative fields of the cortex at 100× magnification were analyzed for each patient slide using Adobe Photoshop 4.0 (Adobe Systems, Inc., San Jose, CA, USA). The interstitial collagen content in the autopsy cohort (‘fibrotic index’) was then calculated as the Sirius Red‒positive interstitial area (in pixels) divided by the total area of each image (in pixels), expressed as a percentage.

In the biopsy cohort, both the endoglin-positive area and the Sirius Red‒positive area in the renal cortex were measured using ImageJ. Glomeruli and large blood vessels were digitally removed from the analysis. The positive area was measured in ten high-power fields per section (for the control group) or in the complete biopsy (for the DN group) and was divided by the total area of each high-power field. 

### 4.6. Immunofluorescence Staining of Biopsy Material

To characterize the cell type in the interstitium that expresses endoglin, 4-µm paraffin-embedded sections of kidney biopsies obtained from three patients with DN and two healthy donor kidneys deemed unsuitable for transplantation were co-stained for endoglin and several cell markers. Sections were first treated with 10 mM Tris/EDTA (pH 9.0) buffer for antigen retrieval. The following primary antibodies were used: Goat anti-human endoglin (1:800, R&D systems), mouse anti-human CD31 (IgG1, clone JC70A, 1:200; Dako, Glostrup, Denmark) to stain endothelial cells, mouse anti-human CD68 (IgG1, clone KP1, 1:2000; Dako) to stain macrophages, mouse anti-human vimentin (IgG2a, clone Vim 3B4, 1:750; Dako) to stain fibroblasts, and mouse anti-human α-SMA (IgG2a, clone 1A4, 1:400; Dako) to stain myofibroblasts. Donkey anti-goat IgG Alexa 546 (Thermo Fisher Scientific, Waltham, MA, USA) and donkey anti-mouse IgG Alexa 488 (Thermo Fisher Scientific) were used as secondary antibodies to visualize the staining. The sections were mounted with ProLong Gold antifade reagent with DAPI (Life Technologies, Carlsbad, CA, USA).

### 4.7. Cell Culture

The human kidney fibroblast cell line TK173 was a kind gift from Prof. Gerhard A. Müller in the Department of Nephrology and Rheumatology, Georg August University, Göttingen, Germany [64]. Fibroblasts were cultured in DMEM/F12 (Gibco Laboratories, Gaithersburg, MD, USA) supplemented with 10% fetal bovine serum (Sigma-Aldrich), and 0.4% penicillin-streptomycin (Gibco Laboratories) at 37 °C in 5% CO2. Where indicated, fibroblasts were transduced with a lentiviral vector expressing either an shRNA against *ENG* mRNA (hereafter referred to as *ENG^KD^* fibroblasts) or non-targeting vector control (hereafter referred to as *ENG^ctrl^* fibroblasts); the lentiviral vectors were kindly provided by Dr. L. Hawinkels in the Department of Gastroenterology, LUMC. 

### 4.8. Analysis of Endoglin Expression 

To determine the efficiency of *ENG* knockdown, total RNA was isolated from *ENG^ctrl^* and *ENG^KD^* fibroblasts one day after reaching confluence using TRIzol (Ambion, Foster City, CA, USA) in accordance with the manufacturer’s instructions. The RNA was converted to cDNA, which was analyzed with qPCR using the IQTM SYBR Green Supermix (Bio-Rad, Hercules, CA, USA) with a CFX real-time system (Bio-Rad, Hercules, CA). Cycle threshold (Ct) values were normalized to the housekeeping gene *HPRT1*. The primers used are shown in Table 2. 

For western blot analysis, one day after reaching confluence *ENG^ctrl^* and *ENG^KD^* fibroblasts were washed in ice-cold phosphate-buffered saline (PBS) and lysed in Tris-buffered saline (TBS) (pH 7.5) containing 1% (w/v) sodium dodecyl sulfate (SDS), 10mM EDTA, and protease and phosphatase inhibitor cocktails (Roche, Basel, Switzerland). Protein concentration was determined using a detergent-compatible protein assay (Bio-Rad). Protein lysates were separated by SDS-PAGE and transferred to membranes for western blot analysis. The membranes were blocked for 1 h in TBS-Tween containing 5% (w/v) bovine serum albumin (BSA), and then incubated overnight with a primary antibody against endoglin (1:1000, R&D Systems); GAPDH (1:3000, Cell Signaling Technology, Danvers, MA, USA) was used as a loading control. IRDye Infrared Fluorescent antibodies (LI-COR Biosciences, Lincoln, NE, USA) were used as the secondary antibody, and the signals were visualized using the LI-COR Odyssey Infrared Imaging System (LI-COR Biosciences). Band intensity of the endoglin western blot was quantified using Odyssey V3.0 software (LI-COR Biosciences). Each experiment was performed in triplicate.

### 4.9. Immunofluorescence Staining of Cell Culture

To stain cultured cells for α-SMA, 2x10^4^
*ENG^KD^* or *ENG^ctrl^* fibroblasts per well were plated on a glass coverslip in a 24-well plate. Two days after plating, fibroblasts were starved in serum-free medium for 8 h and subsequently incubated with 5 ng/mL TGF-β (PeproTech, London, UK) for 24 h. After incubation, fibroblasts were washed in ice-cold PBS and in fixated in 100% ice-cold methanol. As primary antibody, mouse anti-human α-SMA (IgG2a, clone 1A4, 1:500; Dako) was used; Donkey anti-mouse IgG Alexa 488 (Thermo Fisher Scientific) was used as secondary antibody. The sections were mounted with ProLong Gold antifade reagent with DAPI (Life Technologies).

### 4.10. Analysis of TGF-β-Induced Changes in Gene and Protein Expression

To analyze the effect of TGF-β1 stimulation on downstream signaling targets, 10^5^
*ENG^KD^* or *ENG^ctrl^* fibroblasts per well were plated in a 12-well plate. One day after the fibroblasts reached confluence, they were further cultured in serum-free medium and after five hours, subsequently incubated with 5 ng/mL TGF-β1 (PeproTech,) for one hour (pSmad western blots) or for 24 h (qPCR’s and α-SMA western blot). As a control, some cells were incubated with 0.1% (w/v) BSA in PBS. RNA isolation, cDNA synthesis, and qPCR were performed as described above, and Ct values were normalized to the housekeeping gene *HPRT1*. The primers used are shown in Table 2. 

Protein concentration and western blot analysis were performed as described above using rabbit anti-Phospho-Smad1/5/9 (IgG, Ser463/465, 1:800, Cell Signaling), rabbit anti-Phospho-Smad2/3 (IgG, Ser465/467, 1:800, Cell Signaling), mouse anti-α-SMA (IgG2a, Clone 1A4, 1:500, Dako) and GAPDH (1:3000, Cell Signaling Technology). Each experiment was performed in triplicate. Band intensity for the α-SMA western blot was quantified using Odyssey V3.0 software (LI-COR Biosciences). Band intensity for the pSmad western blots was quantified using Image Studio^TM^ Lite Software (LI-COR Biosciences).

### 4.11. Statistical Analysis

The interstitial endoglin-positive area in DN patients and diabetic controls were compared using a Chi-square test for trend. The *p*-values for the putative correlations between the interstitial endoglin-positive area and the characteristics of DN patients in the autopsy cohort were obtained using a Chi-square test for trend (trend of ordered categorical outcomes over-ordered groups), or Spearman’s rank correlation (trend of continuous outcome variables over-ordered groups). The data obtained for the biopsy cohort and cell culture experiments were analyzed using the Student’s *t*-test. A *p*-value <0.05 was considered statistically significant.

## Figures and Tables

**Figure 1 ijms-21-07713-f001:**
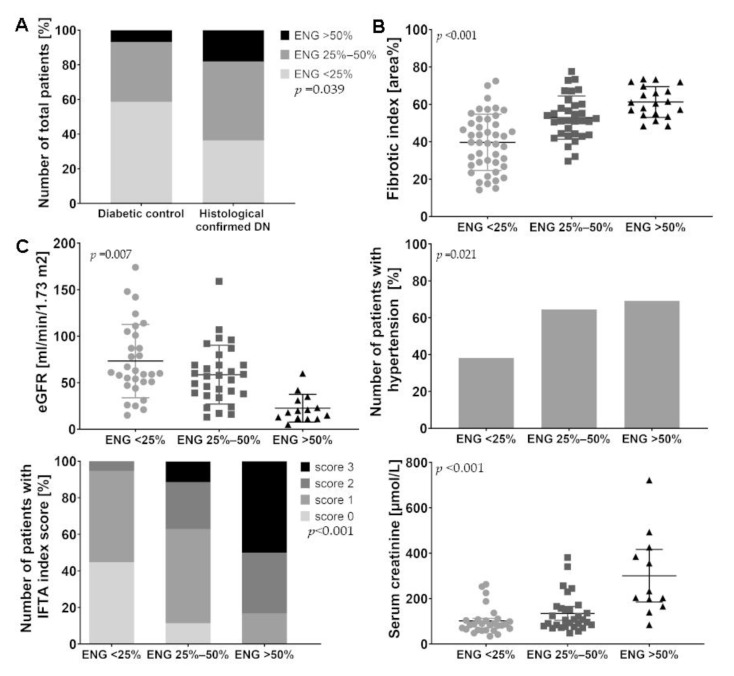
Characterization of the endoglin-positive interstitial area in an autopsy cohort, and correlation with renal damage. (**A**,**B**) Autopsy samples from 55 patients with diabetic nephropathy (DN) and 25 diabetic controls without DN were stained using an endoglin antibody and Sirius Red (to measure interstitial fibrosis). (**A**) Endoglin staining was scored semi-quantitatively, and patients were classified based on their scores. The group with DN had significantly more interstitial endoglin (ENG) compared to the diabetic control group (*p* < 0.05). (**B**) High endoglin levels in the 55 patients with DN were correlated with a higher interstitial fibrosis index (*p* < 0.001). (**C**) High endoglin levels in the interstitium of DN patients were correlated with decreased eGFR (*p* < 0.01), the presence of hypertension (*p* < 0.05), a higher IFTA index score (*p* < 0.001), and increased serum creatinine levels (*p* < 0.001). Data were analyzed using a Chi-square test for trend (linear data compared with ordinal data) or Spearman’s rank correlation (categorical/ordinal data compared with ordinal data). IFTA, interstitial fibrosis, and tubular atrophy.

**Figure 2 ijms-21-07713-f002:**
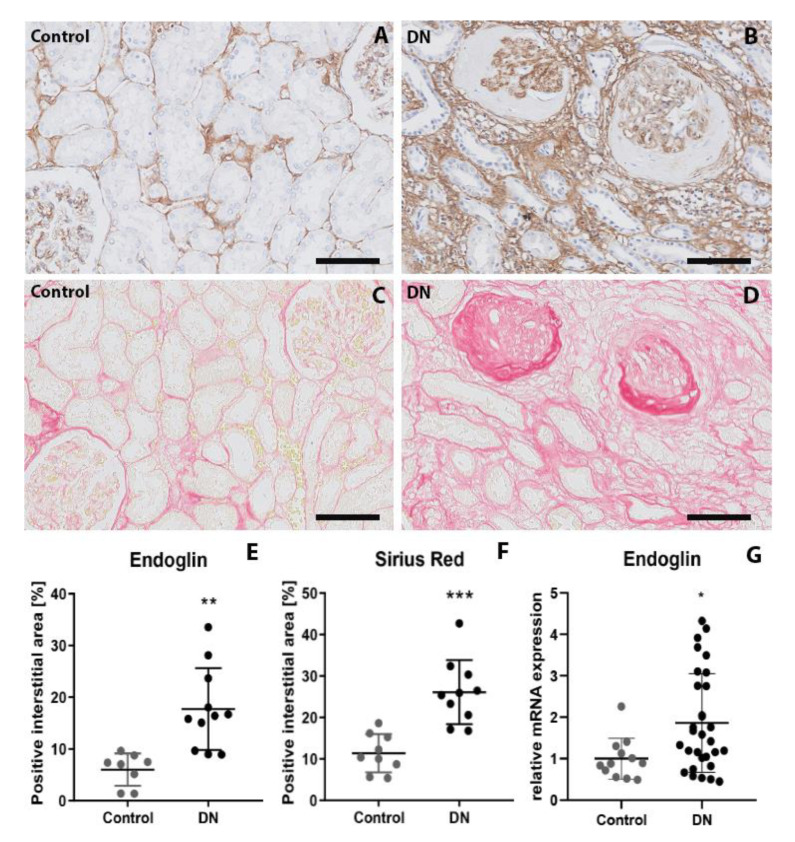
Endoglin expression is increased in the interstitium in biopsies of patients with DN. Representative images of endoglin staining (**A**,**B**) and Sirius Red staining (**C**,**D**) in kidney biopsies from non-diabetic controls (**A**,**C**) and patients with DN (**B**,**D**). Summary data for the endoglin-positive area (**E**) and Sirius Red‒positive area (**F**) are also shown. (**G**) mRNA expression levels of endoglin in whole kidney lysates derived from biopsies of patients with DN. The housekeeping gene *HPRT* was used as a control. * *p* < 0.05, ** *p* < 001 and *** *p* < 0.001 (Student’s *t*-test). The scale bars represent 50 µm.

**Figure 3 ijms-21-07713-f003:**
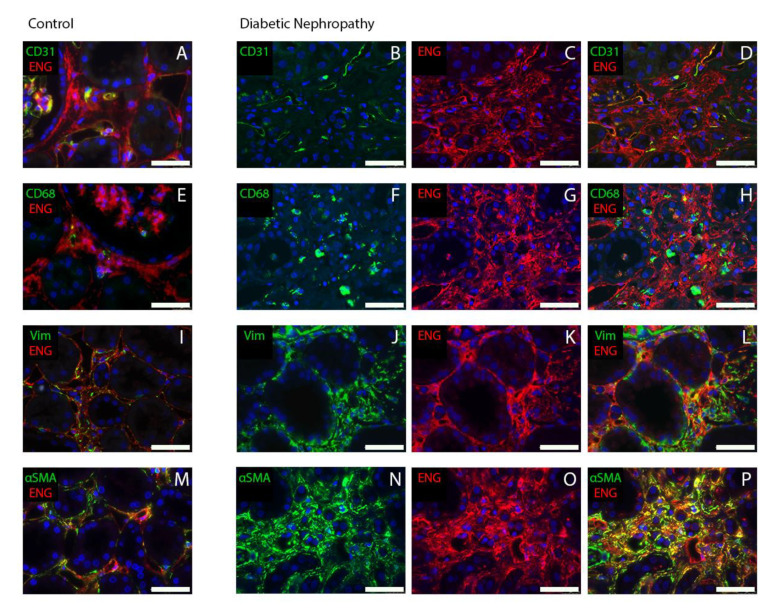
Endoglin is co-localized with the myofibroblast marker α-SMA. Kidney biopsy samples from non-diabetic controls (**A**,**E**,**I**,**M**) and patients with DN (**B**–**D**,**F**–**H**,**J**–**L**,**N**–**P**) were co-immunostained for endoglin (red) and the endothelial cell marker CD31 (**A**–**D**), the macrophage marker CD68 (**E**–**H**), the fibroblast marker vimentin (**I**–**L**), or the myofibroblast marker α-SMA (**M**–**P**). Note the co-localization of endoglin and the endothelial marker CD31 in blood vessels, the co-localization of endoglin and the fibroblast marker vimentin, as well as the co-localization of endoglin and the myofibroblast marker α-SMA, particularly in the patients with DN (**H**). The scale bars represent 50 µm.

**Figure 4 ijms-21-07713-f004:**
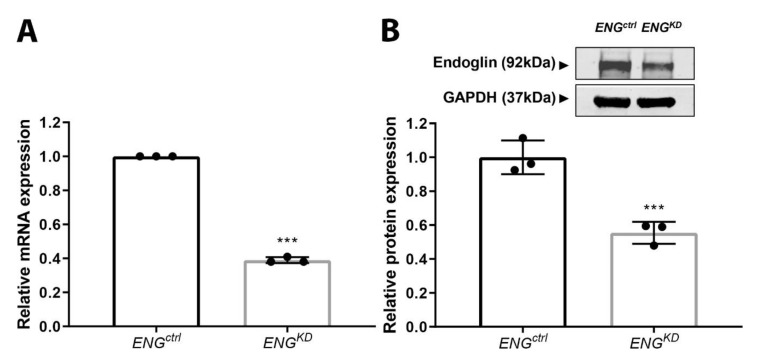
shRNA-mediated knockdown of endoglin expression in TK173 fibroblasts. (**A**) *ENG* mRNA levels were measured in TK173 cells expressing an empty vector control (*ENG^ctrl^*) and TK173 cells expressing an shRNA against *ENG* (*ENG^KD^*); data are depicted relative to *ENG^ctrl^* cells. (**B**) Western blot analysis of endoglin protein levels in *ENG^ctrl^* and *ENG^KD^* fibroblasts. GAPDH was used as a loading control. Data are shown relative to *ENG^ctrl^* cells. Data are presented as the mean ± SD from three experiments. *** *p* < 0.001 (Student’s *t*-test).

**Figure 5 ijms-21-07713-f005:**
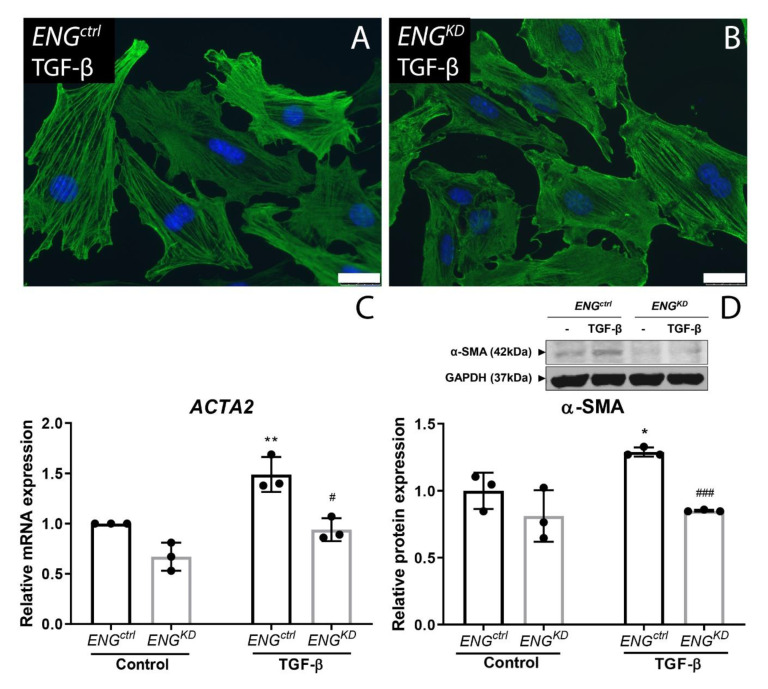
TGF-β-induced upregulation of the myofibroblast differentiation marker α-SMA is reduced in *ENG^KD^* fibroblasts. *ENG^ctrl^* (**A**) and *ENG^KD^* (**B**) fibroblasts were incubated with 5 ng/mL TGF-β for 24 h and stained for α-SMA after fixation in ice-cold 100% methanol. The scale bars represent 25 µm. (**C**,**D**) *ENG^ctrl^* and *ENG^KD^* fibroblasts were incubated with 5 ng/mL TGF-β1 for 24 h, after which *ACTA2* mRNA (**C**) and α-SMA (**D**) were measured using qPCR and western blot analysis, respectively. In (**C**), the housekeeping gene *HPRT* was amplified as a control; in (**D**), GAPDH was used as a loading control. *ACTA2* mRNA and α-SMA protein levels are plotted relative to unstimulated (control) *ENG^ctrl^* fibroblasts, and summary data are presented as the mean ± SD from three experiments. * *p* < 0.05 and ** *p* < 0.01 versus unstimulated (control) *ENG^ctrl^* fibroblasts (Student’s *t*-test); ^#^
*p* < 0.05 and ^###^
*p* < 0.001 versus stimulated *ENG^ctrl^* fibroblasts (Student’s *t*-test).

**Figure 6 ijms-21-07713-f006:**
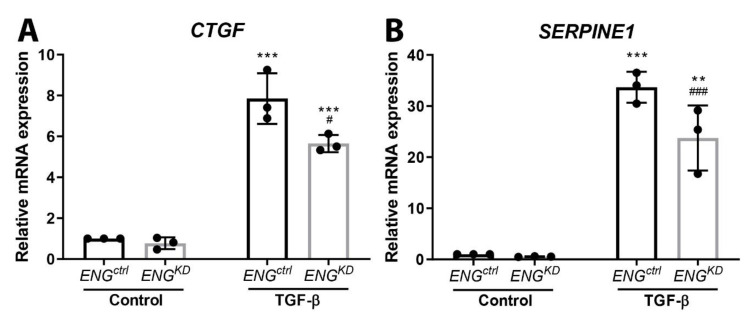
*ENG^KD^* fibroblasts have reduced TGF-β-induced upregulation of pro-fibrotic TGF-β downstream target genes. *ENG^ctrl^* and *ENG^KD^* fibroblasts were incubated with 5 ng/mL TGF-β1 for 24 h, after which *CTGF* (**A**) and *SERPINE1* (**B**) mRNA levels were measured using qPCR. The housekeeping gene *HPRT* was amplified as a control, and each mRNA level is plotted relative to the respective unstimulated (control) *ENG^ctrl^* group. mRNA levels are plotted relative to unstimulated (control) *ENG^crtl^* fibroblasts, and summary data are presented as the mean ± SD from three experiments. ** *p* < 0.01 and *** *p* < 0.001 versus unstimulated fibroblasts from the same cell line (Student’s *t*-test); ^#^
*p* < 0.05 and ^###^
*p* < 0.001 *ENG^ctrl^* versus stimulated *ENG^KD^* fibroblasts (Student’s *t*-test).

**Figure 7 ijms-21-07713-f007:**
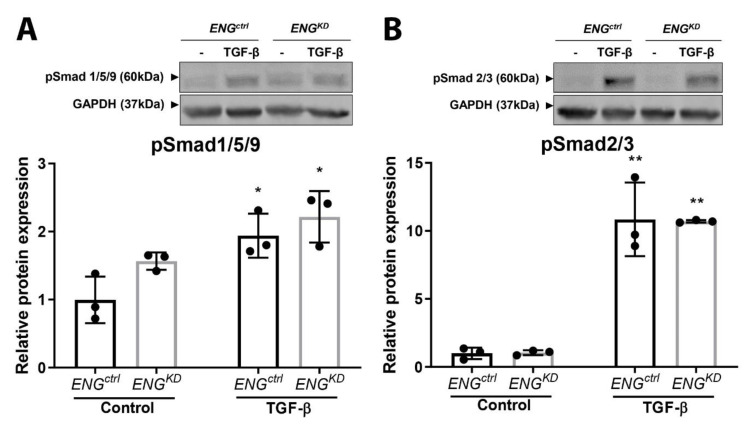
TGF-β-induced Smad1/5/9 and Smad2/3 phosphorylation is similar in *ENG^KD^* and *ENG^ctrl^* fibroblasts. *ENG^ctrl^* and *ENG^KD^* fibroblasts were incubated with 5 ng/mL TGF-β1 for 1 h, after which pSmad1/5/9 (**A**) and pSmad2/3 (**B**) were measured using western blot analysis. GAPDH was used as a loading control. The lower graphs show the quantification of the western blot data. pSmad protein levels were plotted relative to unstimulated *ENG^ctrl^* fibroblasts, and summary data are presented as the mean ± SD from three experiments. * *p* < 0.05 and ** *p* < 0.01 versus unstimulated fibroblasts from the same cell line. (Student’s *t*-test).

**Table 1 ijms-21-07713-t001:** Baseline characteristics of patients.

Characteristic			Histological Confirmed DN (*n* = 55)	Diabetic Controls (*n* = 25)	*p*
Male sex		*n* (%)	34 (61.8)	17 (68.0)	0.594 *
Age years		Mean ± SD	67.4 ± 13.8	67.4 ± 12.7	0.996 ^†^
Type 1 Diabetes		*n* (%)	5 (10.4)	3 (13.0)	0.541 *
eGFR (ml/min/1.73 m^2^)		Mean ± SD	51.7 ± 34.9	72.4 ± 38.7	0.035 ^†^
Serum creatinine (µmol/L)		Mean ± SD	167.3 ± 116.6	103.7 ± 51. 5	0.024 ^†^
Systolic blood pressure (mmHg)		Mean ± SD	140.8 ± 32.2	134.3 ± 28.2	0.509 ^†^
Diastolic blood pressure (mmHg)		Mean ± SD	76.4 ± 11.6	78.2 ± 18.5	0.682 ^†^
HbA1c (%)		Mean ± SD	8.2 ± 2.2	7.1 ± 1.1	0.205 ^†^
Hypertension present		*n* (%)	26 (59.1)	11 (52.4)	0.609 *
IFTA	Index score 0	*n* (%)	9 (16.4)	12 (48.0)	
	Index score 1	*n* (%)	28(50.9)	10 (40.0)	
	Index score 2	*n* (%)	11 (20.0)	3 (12.0)	
	Index score 3	*n* (%)	7 (12.7)	0 (0.0)	0.013 *

* = calculated with a Chi-squared test; ^†^ = calculated with a Student’s *t*-test.

**Table 2 ijms-21-07713-t002:** qPCR primers.

Gene	Forward	Reverse
*ACTA2*	5’-TTCAATGTCCCAGCCATGTA-3’	5’-GAAGGAATAGCCACGCTCAG-3’
*CTGF*	5’-CCTGGTCCAGACCACAGAGT-3’	5’-TGGAGATTTTGGGAGTACGG-3’
*ENG*	5’-CACTAGCCAGGTCTCGAAGG-3’	5’-CTGAGGACCAGAAGCACCTC-3’
*HPRT*	5’-AGATGGTCAAGGTCGCAAGC-3’	5’-TCAAGGGCATATCCTACAACAAAC-3’
*SERPINE1*	5’-ACTGGAAAGGCAACATGACC-3’	5’-TGACAGCTGTGGATGAGGAG-3’

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
