# Peer review of "Endoglin Promotes Myofibroblast Differentiation and Extracellular Matrix Production in Diabetic Nephropathy"

_ijms, 2020, doi:10.3390/ijms21207713_

Round 1
Reviewer 1 Report
Authors aimed to investigate the role of Endoglin in Diabetic nephropathy. Paper nicely demonstrates the importance of endoglin in the profibrotic process in kidney both in patients and in in vitro. It is well written paper conclusions are supported by the data from histological, immunohistochemical analysis, RT-PCR and Western blot.
Major comments:
- I have no major comments to presented manuscript
- Minor comments
- How many histological slides/sections were used for the quantification of endoglin and siris red positive area?
- Authors mentioned that three representative fields of the cortex at 100x magnification. It means from one slide?
Author Response
Hereby we provide our reply to the Review Report

Reviewer 2 Report
In the manuscript entitled 'Endoglin promotes myofibroblast differentiation and ECM production in Diabetic Nephropathy', the authors aim to correlate the levels of endoglin in human samples of diabetic nephropathy (DN) and elucidate a function of endoglin protein: They demonstrate that endoglin is expressed in myofibroblasts (also in endothelial cells -as it co-localizes with CD31 endothelial marker), and it is involved in myofibroblast differentiation (they explain a mechanism based on the regulation of α-SMA marker.
The story of the paper is very interesiting, and provides a new concept of endoglin function in kidney fibrosis. The seniour author has a great expertise in endoglin function in blood vessels and also kidney fibrosis. The work has it merits, but some discussion and adittional work would be necessary for a good publication:
1- Perhaps the most difficult aspect of the mechanism is how to split the different expression of endoglin in both myofibroblasts and endothelial cells, as both populations behave in a different way in DN and kidney fibrosis. While the myofibroblast pupulation is increased in kidney fibrosis conditions, the endothelial cells can suffer vascular rarefaction:
a) Could authors analyze the changes in the levels of CD31 after DN?.
b) Could the authors add and split the analysis of 'interstitial endoglin' in 'myofibroblast-endoglin' and 'interstitial-endoglin'. The results will be very interesting for the analysis.
Moreover, it will be very interesting to focus the discussion a bit the possible role of endothelial endoglin, and the changes that endothelial endoglin can suffer in DN.
2- The second issue that the authors could improve in the manuscript is the regulation of α-SMA marker.
-Could the authors perform immunofluorescence staining to see different expression and α-SMA or different organisation of the αSMA fibers?.
-And, how endoglin regulates TGF-β-induction of α-SMA marker. Is a SMAD-dependent mechanism?.
Apart from these issues that perhaps need attention, the experimental work is nicely shown and the manuscript has its potential for a good publication.
Round 2
Reviewer 2 Report
The authors have paid attention to most of my suggestions and they now provide new data which has made more attractive this research work.
They also updated the discussion of the manuscript based on the new data according to my suggestion,
I will be very happy to see this manuscript published,
All the best